# Host Defense Proteins and Peptides with Lipopolysaccharide-Binding Activity from Marine Invertebrates and Their Therapeutic Potential in Gram-Negative Sepsis

**DOI:** 10.3390/md21110581

**Published:** 2023-11-07

**Authors:** Tamara Fedorovna Solov’eva, Svetlana Ivanovna Bakholdina, Gennadii Alexandrovich Naberezhnykh

**Affiliations:** G.B. Elyakov Pacific Institute of Bioorganic Chemistry, Far-Eastern Branch of the Russian Academy of Science, Vladivostok 690022, Russia; soltaf@mail.ru (T.F.S.); naber1953@mail.ru (G.A.N.)

**Keywords:** lipopolysaccharide (LPS, endotoxin), LPS-binding proteins/peptides, host defense proteins/peptids, innate immune system, marine invertebrates, Gram-negative sepsis, endotoxic shock

## Abstract

Sepsis is a life-threatening complication of an infectious process that results from the excessive and uncontrolled activation of the host’s pro-inflammatory immune response to a pathogen. Lipopolysaccharide (LPS), also known as endotoxin, which is a major component of Gram-negative bacteria’s outer membrane, plays a key role in the development of Gram-negative sepsis and septic shock in humans. To date, no specific and effective drug against sepsis has been developed. This review summarizes data on LPS-binding proteins from marine invertebrates (ILBPs) that inhibit LPS toxic effects and are of interest as potential drugs for sepsis treatment. The structure, physicochemical properties, antimicrobial, and LPS-binding/neutralizing activity of these proteins and their synthetic analogs are considered in detail. Problems that arise during clinical trials of potential anti-endotoxic drugs are discussed.

## 1. Introduction

Lipopolysaccharide (LPS, endotoxin), the major structural component of the Gram-negative bacteria’s outer membrane, serves as a physical barrier that protects bacteria from the external environment and can be released into the surrounding medium during cell division or death [1]. These molecules are the potent stimulators of the innate immune system, playing an important role in the pathogenesis of Gram-negative infections in animals. When it enters the body of a warm-blooded host, LPS binds and activates the cellular Toll-like receptor 4 (TLR4), which leads to the development of an inflammatory reaction and, ultimately, as a result, to the death and elimination of the pathogen. However, the accumulation of endotoxin in the bloodstream in large quantities (with Gram-negative infection, invasive procedures, etc.) can cause the excessive activation of immunocompetent cells, inducing the overproduction of pro-inflammatory cytokines and a systemic inflammatory response, which leads to sepsis and septic shock within a few days (Figure 1). Lipid A, the predominantly lipophilic and most conserved fragment of the LPS molecule, directly interacts with TLR4 and is responsible for most of the immunobiological and toxic properties of endotoxin [1].

Gram-negative sepsis remains a serious unresolved problem in clinical medicine. This type (according to the etiology of pathogens) of sepsis is clinically the most severe, often accompanied by septic shock, and has a significantly higher mortality rate than other types. The increasing use of new technologies in medical practice—cytostatic and immunosuppressive therapy and transplantation and prosthetics—as well as the HIV and COVID-19 pandemics and the increasing resistance of pathogens to antibiotics contribute to the growth of septic complications in patients. The aging population, accompanied by an increase in chronic diseases, also leads to an increase in the incidence of sepsis and septic shock. As a result, despite the great advances in antimicrobial chemotherapy, mortality rates from septic shock remain the highest and do not decrease worldwide. The development and introduction of new antibiotics into medical practice does not fundamentally solve the problem; moreover, their use in some cases can lead to a deterioration in the condition of a septic patient.

Modern medicine does not have specific and effective anti-sepsis drugs whose molecular target is LPS. Molecules that can selectively block TLR4, preventing endotoxin from binding to receptor and the development of a systemic inflammatory response, may have therapeutic potential for the treatment of sepsis. In particular, TLR-4 receptor antagonists are structural analogs of lipid A with low toxicity, including native lipid A from a number of marine proteobacteria (Proteobacteria) [2,3]. Another approach to the endotoxin-neutralizing drug design is based on the use of synthetic or natural substances that can suppress the biological activity of LPS due to the formation of strong complexes with it (Figure 1) [4]. Searching for such compounds among the host defense proteins of marine invertebrates—which represent one of the most extensive and diverse groups of animals, numbering 153,434 species—seems very promising (UN data for 2019).

Marine invertebrates only have an innate immune system, including a huge set of defense proteins, which was formed during a long evolution and allowed them to survive in the environment enriched with pathogenic microorganisms [5]. Host defense proteins are traditionally referred to as antimicrobial peptides, although they have been shown to be polyfunctional compounds. These proteins are constitutively expressed and rapidly induced in various cells and tissues, interact directly with infectious agents, and/or activate immune responses to eliminate pathogens. Defense proteins usually recognize and bind to the surface of the pathogen the most conservative biopolymers that are common and vital for a large group of microorganisms but not present in the host. These molecules, known as pathogen-associated molecular patterns (PAMPs), trigger innate immune responses in the host [6]. In Gram-negative bacteria, this PAMP is LPS.

Host defense proteins and peptides with lipopolysaccharide-binding capacity from marine invertebrates (ILBPs, invertebrate lipopolysaccharide-binding proteins) may possess different biological properties depending on the structure and nature of the interaction with endotoxins. With a high affinity for LPS, these proteins may have antimicrobial or endotoxin-neutralizing activity, or both [4]. ILBPs that combine both of these properties are considered today as the most effective potential drugs for the treatment of human sepsis.

This review summarizes data on host defense proteins (antimicrobial peptides) of marine invertebrates that can recognize and bind LPS. Particular attention is focused on the group of ILBPs and their synthetic analogs, which neutralizes endotoxin. The structure, physicochemical properties, LPS-binding/neutralizing activities, and antimicrobial activities of these proteins are considered in detail as far as the sources for their isolation among marine invertebrates. The structural basis of the endotoxin-neutralizing action of ILBPs is discussed.

## 2. Anti-Lipopolysaccharide Factor (ALF)

A cationic protein that inhibited the LPS-induced activation of the crab hemolymph coagulation system has been found in hemocyte lysates of Japanese (*Tachypleus tridentatus*) and American (*Limulus polyphemus*) horseshoe crabs [7,8,9]. This protein, called the anti-lipopolysaccharide factor (ALF), was able to bind LPS, neutralize its biological activity (in vitro and in vivo), and inhibit the growth of R-type Gram-negative bacteria. The natural and recombinant anti-LPS factor Limulus (ALF-L) also suppressed endotoxin-mediated activation of cultured endothelial and B cells, reduced the concentration of endotoxin and TNF-α in the blood serum of experimental animals, and protected them from death in the late stages of endotoxemia and sepsis [10,11,12,13,14].

Numerous ALF homologues have been identified and characterized in different types of crustaceans: shrimp, lobster, crabs, and crayfish [15,16,17,18,19,20,21,22]. The most extensively studied ALFs are isolated from shrimp of the Penaeidae family, which includes many economically important species that are of interest as objects of industrial fishing and breeding [23].

ALFs show a broad spectrum of antimicrobial activity against Gram-negative and Gram-positive bacteria, fungi, human enveloped viruses (herpesvirus type 1, adenovirus), and white spot syndrome virus (WSSV), which is widely distributed throughout the world and considered as one of the most destructive and pathogenic viruses in shrimp farms [24,25,26].

In crustaceans, one organism usually contains several isoforms of ALF, which are either encoded by different genes or formed as a result of alternative mRNA splicing [27]. Thus, six isoforms were identified in the tiger shrimp *Penaeus monodon*, and seven isoforms were found in the Chinese shrimp *Fenneropenaeus chinensis* and the South Korean blue crab *Portunus trituberculatus* [15,28,29,30,31,32]. The isoforms differed in tissue distribution and antimicrobial properties. The wide diversity of ALF sequences within a species may provide a synergistic enhancement of their protective action against bacterial infection.

ALFs are a group of small single-domain antimicrobial proteins consisting of 97–124 amino acid residues with a relatively short 16–28 residue signal sequence. The molecular weight of the mature protein is about 11 kDa: ALFs from *L. polyphemus* and shrimp of various species have masses of 11.8 and 10.74 to 12.23 kDa, respectively [9,33]. According to the values of the isoelectric points (pI), ALFs were classified as cationic peptides, but more and more data are emerging on the existence of anionic proteins among them [33,34,35]. The theoretical pI values of mature shrimp ALFs range from 5.02 to 10.29 [33]. Typically, ALF molecules have a highly hydrophobic N-terminal region and conserved cluster of positively charged and hydrophobic amino acid residues within a loop fixed by a disulfide bond between two conserved cysteine residues, which is commonly referred to as the LPS-binding domain [36]. This amphipathic loop is an important functional molecule moiety, which is responsible for the biological activity of ALFs. Indeed, synthetic peptides corresponding to this fragment from various ALFs have shown antimicrobial activity, the ability to inhibit WSSV virus replication, and a protective effect in sepsis [20,37,38,39].

Despite the large number of registered ALFs (more than 300 proteins of this class from crustaceans were isolated and characterized until 2021), only one crystal structure of them has been established to date. The X-ray structure of recombinant ALF-L consists of three α-helices (one at the N-terminus and two at the C-terminus) packed against a four-stranded β-sheet, giving rise to a wedge-shaped molecule [36]. The potential LPS-binding domain includes an amphipathic β-hairpin formed by the longest β-strands S2 and S3 of the β-sheet and stabilized by the single disulfide bond (Cys31–Cys52). The positively charged residues within the β-hairpin of ALF-L are supposed to interact with the negatively charged phosphate groups of lipid A. However, the lipid A binding site on ALF remains poorly understood to date. Later, the spatial ALF structure from the shrimp *P. monodon*, expressed in yeast cells, rALFPm3, was determined using NMR (Figure 2) [40]. The structure of rALF-Pm3, like ALF-L, is composed of three α-helices, a four-stranded β-sheet, and contains a β-hairpin formed by S2 and S3 β-strands closely linked via a Cys34-Cys55 disulfide bond. A Comparison of the 3D structures of these proteins revealed highly similar clusters of positively charged and hydrophobic residues on the β-sheet surface. This suggests that ALF-L and ALFPm3 have a similar LPS binding site, which is located on the β-sheet and mainly consists of 5–6 positively charged and several hydrophobic residues capable of binding lipid A through electrostatic and hydrophobic interactions.

A series of peptides of various lengths, including cyclic ones, derived from the ALF-L sequence, were synthesized [41]. These peptides demonstrated high endotoxin-binding and neutralizing activities, comparable with those of the parent recombinant protein, and were non-toxic for erythrocytes or cultured human monocytes. A new class of peptides based on the LPS-binding domain of ALF-L or part of it with significant changes in length and primary sequence calculated for optimal lipid A binding was also designed [42]. A preclinical study revealed that these peptides have high selectivity for LPS, as well as high LPS-neutralizing activity in vitro and the ability to protect against sepsis in vivo. An analysis of the obtained data showed that the endotoxin-neutralizing activity of the peptides is closely related to their affinity for LPS and the ability to incorporate into LPS aggregates with changes in their structure. The authors highly appreciate the potential of synthetic peptides as drugs for the treatment of endotoxemia and sepsis.

## 3. β-Hairpin Peptides

### 3.1. Arenicins

Several antimicrobial peptides were isolated from the coelomocytes of marine polychaeta lugworm *Arenicola marina* called arenicins-1, -2, and -3 [43,44]. Arenicin molecules consist of 21 amino acid residues and have the amphipathic β-hairpin structure, formed by the two-stranded antiparallel β-sheet stabilized by one (arenicin-1,-2) or two disulfide (arenicin-3) bridges (Figure 3a).

Conformational analysis via NMR spectroscopy revealed that the β-sheet in arenicins had a marked right-handed twist in an aqueous solution. This distortion effectively shields the hydrophobic side of the β-sheet from contacts with polar solvent, thus reducing the peptide surface amphipathicity [45,46]. When interacting with membranes or in membrane-mimetic environments, arenicins, form dimers stabilized by hydrogen bonds between parallel N-terminal β-strands in two neighboring molecules. Dimerization induces a substantial conformational change so that the molecules adopt almost planar amphipathic β-sheet structures [47,48]. A significant decrease in the twist of the β-hairpin as a result of arenicin dimerization leads to an increase in its amphiphilicity and stability. Natural arenicins exhibit pronounced antimicrobial activity against a broad spectrum of Gram-positive and Gram-negative bacteria, pathogenic fungi, and yeasts even under high-ionic-strength conditions [43,49]. Mode-of-action studies strongly suggest that the antibacterial activity of arenicins is consistent with their ability to disrupt the integrity of bacterial membranes.

More recently, arenicin-3, a member of the arenicin family, was discovered by the pharmaceutical company Adenium Biotech ApS, which develops novel antibiotics for the treatment of Gram-negative bacterial infections [44]. This peptide is very attractive due to its potent broad-spectrum antibacterial activity, even against multidrug-resistant clinical isolates, and its ability to bind LPS. However, arenicin-3 is toxic to mammalian cells and causes hemolysis of human erythrocytes. To solve the problem of toxicity, the structural analogs of arenicin-3 were designed by changing the number of disulfide bonds, hydrophobicity, or charge of the molecule [50,51]. As a result, peptides were synthesized with low cytotoxicity while maintaining antibacterial properties. The optimized synthetic arenicin-3 derivatives, which retained the β-hairpin structure stabilized by one or two disulfide bonds, were found to be the most active against Gram-negative bacteria. These peptides have demonstrated the ability to bind LPS with higher affinity than polymyxin B and neutralize its toxic effects. Also, these synthetic analogs increased the survival of mice during LPS-induced peritonitis and sepsis, protected from lethal LPS challenge in vivo, endotoxin-induced lung injury, and death caused by bacterial infection (*E. coli* and *S. enteritidis*) and also inhibited the production of proinflammatory cytokines. At the same time, they had low hemolytic activity and cytotoxicity, as well as higher antimicrobial activity than the natural peptide. According to the researchers, these optimized arenicin-3 analogs may be potential candidates for the creation of dual-acting drugs with antibacterial and anti-endotoxin activities. Currently, one of the obtained variants is undergoing preclinical trials [52].

### 3.2. Tachyplesins and Polyphemusins

In horseshoe crab hemocytes, in addition to ALF, another group of antimicrobial LPS-binding peptides was found. These relatively short and structurally closely related peptides, known as polyphemusins I and II and tachyplesins I–III, were isolated from *L. polyphemus* and *T. tridentatus*, *Tachypleus gigas*, and *Carcinoscorpius rotundicauda*, respectively [53,54,55].

Their concentration in hemocytes is extremely high, up to 10 mg in the total hemolymph of an individual horseshoe crab [56]. These peptides are 17–18 amino acid residues in length and have an amidated C-terminal arginine residue and a net positive charge. The spatial structure of the peptides in aqueous solutions is an amphiphilic, antiparallel β-hairpin connected by a β-turn and stabilized by two disulfide bonds (Figure 3b,c) [57,58,59,60]. The peptide structure is highly stable and is preserved when samples are heated to 100 °C in neutral pH buffers and kept at low pH [55]. This stability seems to be due to the rigid structure imposed by the two disulfide linkages. In the presence of dodecylphosphocholine micelles, conformational changes in tachyplesin structure occur, which are accompanied by an increase in the amphiphilicity of the molecule and the formation of a contiguous well-defined hydrophobic surface [58]. The ability of peptides to adopt distinct conformations in solution and upon membrane association appears to be partly responsible for their wide range of biological activities, including antimicrobial, antitumor, and anti-inflammatory. So far, for these peptides, the relationships between their structures and functions are not well understood.

These peptides recognize and bind LPS and quickly integrate into LPS monolayers. They are able to displace divalent cations from their binding sites with LPS and penetrate into the outer membrane of bacteria and also demonstrate resistance to inhibition from these cations of incorporation into LPS monolayers [61,62,63]. The recognition site for the peptide on the LPS molecule is the lipid A moiety. The peptides were found to show higher (280-fold for tachyplesin I) affinity for LPS compared with acidic phospholipids. When tachyplesin interacts with LPS, slight changes in its secondary structure are observed: the β-sheet is elongated and twisted, and the whole structure is stabilized. A twisting β-sheet structure may be important for tachyplesin to recognize LPS. According to the proposed model of the complex of tachyplesin I with LPS, the peptide lies across two D-glucosamine residues of lipid A, and its cationic and aromatic residues interact with phosphate groups and acyl chains of lipid A moiety, respectively [62].

The binding of peptides to LPS is accompanied by the neutralization of its toxic effect on the macroorganism. Polyphemusins inhibit the production of pro-inflammatory cytokines TNF-α and IL-6 through LPS-stimulated macrophages, protect mice from endotoxemia, and block the development of endotoxin shock in an animal model. The structural analog study of polyphemusins and tachyplesins made it possible to establish that the antiendotoxin activity of peptides increases with an increase in their binding affinity to LPS and the amphiphilicity of the molecule [63].

Tachyplesins and polyphemusins show pronounced activity against a wide range of microorganisms and, along with arenicins, are considered the most active antimicrobial peptides of animal origin. They inhibit the growth of both Gram-positive and Gram-negative bacteria, as well as some fungi at sub-micromolar and micromolar concentrations [53,63,64,65].

Despite the commonality of the beta-hairpin fold stabilized by disulfide bonds and a wide spectrum of activity against both Gram-negative and Gram-positive bacteria, tachyplesins/polyphemusins and arenicins have a rather low degree of amino acid sequence similarity (up to 35%) [66] and differ in the mechanism of antimicrobial action. Tachyplesins and polyphemusins have been shown to translocate across membranes without significant disruption of lipid bilayers [67], while arenicins disrupt the cell membrane through the formation of higher oligomeric states [46].

## 4. Big Defensins

Big defensins were first discovered in horseshoe crabs. A novel defensin-like protein was isolated from *T. tridentatus* hemocytes, which contained 79 amino acid residues and was named “big defensin” (BigDef). This protein had a pronounced ability to bind LPS, as well as to inhibit the growth of Gram-positive and Gram-negative bacteria and fungi [68]. The BigDef molecule consists of a highly hydrophobic N-terminal domain and a cationic C-terminal domain containing six cysteine residues involved in three internal disulfide bridges. These two different domains are connected via a flexible linker. The spatial structure of the C-terminal domain is a twisted three-stranded antiparallel β-sheet, stabilized by three disulfide bonds, and the N-terminal domain adopts a conformation formed by parallel β-sheet and two α-helices, which in the lipid environment are transformed into an elongated single α-helix (Figure 4a) [69,70].

The C-terminal domain is structurally similar to human β-defensins, HβD-2 and HβD-3 [71] and differs from invertebrate defensins. Trypsin cleaves BigDef at the Arg-37 residue to form two peptide fragments that have diverse activities. The N-terminal hydrophobic peptide acts predominantly against Gram-positive bacteria, while the C-terminal cationic peptide is more active against Gram-negative bacteria. Both generated peptides showed weak LPS-binding activity, whereas the activity of intact full-length BigDef was significant compared with that of anti-LPS factor ALF peptide from *T. tridentatus* hemocytes. Thus, binding to LPS requires the native conformation of the entire molecule.

Phylogenetic analysis of all currently known sequences of the BigDef genes showed that these proteins form a group predominantly represented in marine invertebrates, mainly in mollusks [72,73,74] and, to a much lesser extent, in horseshoe crabs and lancelets [75]. The spatial structure of the big defensin from the Pacific oyster *Crassostrea gigas* (Cg-BigDef1) was recently determined (Figure 4b) [76]. This is the second currently known structure for a protein from the BigDef family. NMR spectroscopy revealed that oyster Cg-BigDef1, like horseshoe crab Tt-BigDef, possesses two structural domains. Cg-BigDef differs from Tt-BigDef in the orientation of the N- and C-terminal domains, the length of the linker sequence, and, as a result, surface properties. The big defensin overall structure from the oyster is mainly hydrophobic, while that from the horseshoe crab is amphiphilic. This suggests that Cg-BigDef1 binding to bacterial membranes occurs through hydrophobic rather than electrostatic interactions and is not impaired at high salt concentrations. Cg-BigDef1 exhibited salt-stable activity against both Gram-positive and Gram-negative bacteria and fungi. Genes encoding homologous proteins from two other mollusk species, *Venerupis philippinarum* and *Argopecten irradians*, have been cloned and expressed, and the obtained recombinant proteins have been characterized [77,78]. These molecules are cationic peptides with a molecular weight of 8–11 kDa and an isoelectric point of 8.6–9.2. They exhibit antibacterial and antifungal activities. The ability of these proteins to bind LPS has not been studied.

## 5. Factor C

Factor C is a unique LPS-binding protein found in horseshoe crabs. It is a trypsin-like serine protease zymogen. The zymogen is activated by picogram amounts of LPS and is the initiator of the coagulation cascade, which the horseshoe crab uses as one of the defense mechanisms against pathogens [79]. It can specifically bind LPS on the surface of hemocytes and directly recognize Gram-negative bacteria in an LPS-dependent manner, acting as a pattern-recognizing receptor [80]. Recombinant Factor C (rFC) interacts at extremely high affinity with LPS and lipid A: the dissociation constant (K_D_) of its complex with lipid A is 7.6 × 10^−10^ M [81]. Due to high specificity and sensitivity to LPS, it is widely used by pharmaceutical companies to detect endotoxin contamination of parenteral drug products and medical devices (Limulus Amoebocyte Lysate (LAL) test and rFC test) [82]. Recombinant factor C has been shown to effectively inhibit LPS-induced production of TNF-α and IL-8 via human macrophages and is therefore a potential LPS-neutralizing agent [83]. Furthermore, rFC is non-toxic to human monocytes and *HeLa* cells. In horseshoe crab amoebocyte lysate, Factor C was found in two molecular species: in a single-chain form and a two-chain form, which consists of heavy and light chains linked by (a) disulfide bond(s) [84]. Both molecules have the same molecular mass. A single-chain form is transformed into a two-chain form as a result of the LPS-mediated activation of Factor C [85,86]. It is believed that molecular reorganization may occur during the isolation and purification of Factor C due to the presence of LPS trace impurities.

Factor C is a glycoprotein and, depending on the source of isolation, *L. polyphemus* and *T. tridentatus hemocytes* or *Carcinoscorpius rotundicauda* hemocytes, has a molecular weight of 120 or 132 kDa with heavy and light chain sizes of 80 and 43 or 80 and 52 kDa, respectively [86,87]. The Factor C molecule consists of domains that are structurally related to proteins of the mammalian complement system [88]. Along with the typical serine protease domain at the C-terminus, Factor C also includes a cysteine-rich (Cys) region; a domain homologous to epidermal growth factor (EGF); five complement control protein (CCP) modules, also known as Sushi domains; an LCCL segment (a fragment common to a number of proteins such as Coch-5b2 and Lg11 [89]); and a domain similar to a C-type lectin (Figure 5).

The CCP module consists of approximately 60 amino acid residues, including four cysteine residues, forming two internal conserved disulfide bonds, and has a β-sandwich spatial structure with a compact hydrophobic core. One face of the β-sandwich is made up of three β-strands linked by hydrogen bonds, and the other face formed by two separate β-strands [90]. The regions of the polypeptide chain between the β-strands are composed of both well-defined turns and less well-defined loops. An analysis of CCP sequence alignments reveals a high degree of conservation among residues of obvious structural importance, while almost all insertions, deletions, or substitutions are found in the region of the loops. This suggests that the structure of the 16th CCP module from human complement factor H (Figure 6a) gives sufficient understanding of the structure of these modules in the Factor C molecule.

The EGF-like domains are about 50 amino acid residues in length and contain six cysteine residues that form disulfide bonds, resulting in a very compact configuration [91,92]. The secondary structure of these polypeptides in water contains two antiparallel β-sheets and several β-turns. Figure 6b shows the crystal structure of human EGF (hEGF), which consists of an N region and a C region [93]. The N region has an irregular N-terminal peptide segment and an anti-parallel β-sheet. The C region contains a short anti-parallel β-sheet and a C-terminal segment, which are probably disordered in isolation. There are two hEGF molecules in the asymmetric unit of the crystals, which form a potential dimer.

The N-terminal fragment of Factor C is completely responsible for binding to LPS [94,95]. This fragment, which includes a Cys-rich region, an EGF-like domain, and three CCP (Sushi) modules, has several LPS (lipid A) binding sites and demonstrates strong positive cooperativity of binding to the ligand [96]. This, apparently, determines its significantly higher ability to neutralize endotoxin, in comparison with polymyxin B, as well as the high sensitivity of Factor C to LPS. At low concentrations, the Factor C fragment completely inhibits LPS-induced production of TNF-α and IL-8 by human monocyte cells THP-1 and peripheral blood mononuclear cells. N-terminal fragment, which has low cytotoxicity, protects mice from LPS-induced lethality.

Structural and functional analysis of the N-terminal region of factor C from the horseshoe crab *C. rotundicauda* showed that LPS-recognizing regions are localized in the CCP 1 (Sushi 1) and CCP 3 (Sushi 3) modules [97]. These modules have high-affinity LPS binding sites with K_D_ from 10^−9^ to 10^−10^ M, which are located in two 34-mer peptides, S1 and S3. Both S1 and S3 can inhibit the LAL reaction with endotoxin and the LPS-induced production of TNF-α by human macrophages with different efficiencies. The analysis of the obtained data showed that at least two S1 peptides cooperatively bind to one LPS molecule with a Hill coefficient of 2.42. In contrast, the binding of LPS to S3 is non-cooperative. Synthetic peptides developed on the basis of CCP modules inhibited the LPS-induced secretion of TNF-α by human THP-1 cells and protected D-galactosamine-sensitized mice from a lethal dose of *E. coli* LPS [96,97], demonstrating the promise of their use for the immunotherapy of Gram-negative sepsis.

In the case of the Factor C orthologue from *T. tridentatus*, the LPS-binding region of the molecule was found not inside the tandem CCP (Sushi) modules but in the N-terminal cysteine-rich Cys region and the EGF-like domain [80]. The Cys-rich region specifically binds LPS not only in free form but also on the bacterial surface. The LPS binding site in the Cys region contains a conserved tripeptide sequence (Arg36-Trp37-Arg38) consisting of an aromatic residue flanked by two basic residues, which is also found in other LPS-recognizing proteins [98]. Mutations in this tripeptide prevent its binding to both LPS and Gram-negative bacteria, which determines the key role of this conserved motif in interaction with LPS. It is assumed that the binding of this peptide to LPS occurs according to the mechanism previously proposed for the ALF peptide: the basic residues interact with D-glucosamine-1-phosphate of lipid A, and the aromatic residue associates with its hydrophobic part.

Full-length Factor C binds and neutralizes LPS more effectively than individual LPS-binding peptides derived from it [83,97]. This fact indicates that interdomain interactions in the molecule enhance the overall interaction between factor C and LPS. The tandem arrangement of Sushi’s LPS-binding domains in Factor C has been reported to be responsible for its high affinity for LPS. In addition, the lectin-like and CCP 4 domains have been shown to contribute to the binding of factor C to LPS [99]. These modules can either influence the conformation of LPS-binding domains or directly participate in LPS binding.

## 6. Bactericidal/Permeability-Increasing Proteins

The bactericidal/permeability-increasing protein (BPI) is a member of the LBP/BPI family of LPS-binding proteins and a component of the innate immune system that acts selectively against Gram-negative bacteria [100]. These proteins have a direct cytotoxic and opsonizing effect on bacteria and also bind LPS in the lipid A region and neutralize its biological activities [101,102,103]. These properties of BPI have been used therapeutically for endotoxin-related complications of various diseases [104]. In animal models of sepsis, pneumonia, and endotoxemia, as well as in preclinical and clinical trials, recombinant BPI peptides have been shown to neutralize many of the biological effects of LPS.

The first BPIs were isolated from rabbit and human polymorphonuclear leukocytes [105,106]. Human BPI (hBPI), one of the most well-studied representatives of the LBP/BPI family, is a cationic protein with a molecular weight of 55 kDa, whose three-dimensional structure was determined by X-ray diffraction (Figure 7) [107,108]. At the same time, orthologues of this protein have been found in several non-mammalian vertebrate species and various invertebrates [109].

The hBPI homologue was revealed in the oyster *C. gigas* [110]. This was the first time that BPI was identified in an invertebrate. The protein, named Cg-BPI, was obtained in recombinant form and characterized. Mature Cg-BPI is a cationic protein (50.1 kDa) with a calculated pI of 9.3, close to that of hBPI, pI 9.4. Amino acid sequence analysis, as well as structure modeling and electrostatic surface potential prediction, showed that Cg-BPI has a high degree of structural similarity to hBPI. It contains two conserved domains, N-terminal and C-terminal, which are separated by a proline-rich region. Although these domains have a low degree of amino acid sequence similarity, they exhibit the same spatial structure. The Cg-BPI, like the hBPI, has a boomerang shape and consists of two identical barrels formed by a beta-sheet and two alpha-helices, which are connected via a central β-sheet (Figure 7). The N-terminal domain of Cg-BPI contains functional regions previously characterized in hBPI as responsible for LPS binding and neutralization, as is bactericidal activity [111,112]. This domain has three conserved lysine residues that can bind to negatively charged LPS groups through electrostatic interactions [113], as well as two cysteines characteristic of mammalian BPI and forming a disulfide bond, which has been shown to be important for the rhBPI function [114]. Both domains contain an apolar pocket that serves as a binding site for lipids and probably lipid A.

Recombinant Cg-BPI bound LPS and lipid A with high affinity (K_D_ 3.1 × 10^−8^ M for *E. coli* LPS) [110]. It had a strong bactericidal effect on Gram-negative bacteria and disrupted their cytoplasmic membranes. Thus, the BPI protein from *C. gigas* combines LPS-binding activity with antibacterial and membrane-permeabilizing properties. It is noteworthy that Cg-BPI was highly active against *E. coli* SBS363, which contains LPS with short O-polysaccharide chains but was 30 times less active against *E. coli* ML35 with long-chain LPS. A similar result was obtained for hBPI and was explained by the greater accessibility of anionic and hydrophobic sites in and near the lipid A region of the LPS molecule in *E. coli* SBS363 due to a decrease in the shielding effect of O-polysaccharide chains [115,116].

In further research, a second BPI, Cg-BPI2, was found in *C. gigas*, which showed the highest sequence identity with the already-known Cg-BPI [117]. According to the results of molecular modeling, Cg-BPI2, like hBPI and Cg-BPI, has a structure consisting of an N- and C-terminal barrels and a central β-sheet. At the same time, a comparison of the electrostatic surface potentials revealed that Cg-BPI2 has a higher surface charge than hBPI and Cg-BPI. The recombinant N-terminal domain of Cg-BPI2 exhibited a high affinity for LPS and was effective against Gram-negative bacteria. Thus, the antibacterial activity of *C. gigas* BPIs, as well as human BPI, is determined by their N-terminal domain.

Recently, BPIs EsBPI2 and EsBPI4 from the squid *Euprymna scolopes* have been characterized [118]. Based on amino acid sequence analysis and comparative modeling data, EsBPI2/4 were predicted to have molecular characteristics typical of hBPI. These two proteins have a two-domain “boomerang-like” structure. They share with other BPIs the predicted LPS-binding regions in their N-terminal domains and conserved cysteines, which are involved in the formation of disulfide bonds crucial for the functional activity of this family of proteins. Both proteins isolated from squid tissue extract showed potent bactericidal activity against Gram-negative bacteria *Vibrio fischeri*. Host exposure to LPS derivatives (lipid A) led to increased EsBPI2 gene expression.

Using genomic technologies, proteins of the LBP/BPI family have been identified in a number of marine invertebrates, such as marine annelids [119], sea urchins [120], and mollusks [121]. The LBP/BPI gene expression in the invertebrates has been shown to occur after challenge with LPS. These results suggest that BPIs contribute to the elimination of Gram-negative bacteria through interaction with LPS.

## 7. Lipopolysaccharide-Binding Lectins

Lectins are non-immunoglobulin-type proteins or glycoproteins that selectively recognize and reversibly bind to specific carbohydrates and carbohydrate moieties without changing the structure of glycan. Marine animals, including invertebrates, have a large and complex set of lectins that vary considerably in their structure and carbohydrate specificity. Invertebrate lectins are potential molecules involved in the immune recognition and phagocytosis of microorganisms through opsonization. They are able to interact with LPS on the surface of bacterial cells. However, this fact only applies to those LPS that have carbohydrate motifs recognized by the lectins. Carbohydrate ligands in the LPS molecule are localized mainly in O-specific polysaccharide chains (hypervariable structural element) and rather less frequently in the core oligosaccharide (relatively conserved structure) and in lipid A (very conserved part of the molecule) [122].

In marine invertebrates, horseshoe crab LPS-binding lectins are the best studied. Five types of such lectins have been isolated from the hemolymph of *T. tridentatus*, of which four, called tachylectins (TL-1 to TL-4), are from hemocytes and one, TPL2 (Tachypleus plasma lectin 2), is from plasma. Unlike tachylectin proteins-1-3, TPL-2 is a glycoprotein. A study of recombinant TPL2 with a mutation in the glycosylation site shows that glycosylation does not appear to be important for LPS binding [123].The hemolymph lectins differ in their carbohydrate specificity. Indeed, TL-1 (L6) interacts with the core oligosaccharide of the LPS molecule, probably through the 2-keto-3-deoxyoctonic acid (KDO) residue [124]; TL-2 (L10) exhibits specific activity for D-GlcNAc (K_D_ = 5.13 × 10^−5^ M and 1.54 × 10^−8^ M for free and immobilized (GlcNAc–BSA) monosaccharide, respectively) and D-GalNAc [125,126]; TL-3 specifically binds to S-type LPS from several Gram-negative bacteria through a specific structural fragment of O-polysaccharide, similar to that of the blood group A antigen [127]; and TL-4, like TL-3, specifically recognizes S-type LPS, but not R-LPS lacking O-polysaccharide. The most likely specific ligand for TL-4 is colitose (3-deoxy-L-fucose), a monosaccharide that is structurally similar to L-fucose, to which the lectin is also able to bind, but with lower affinity [128]. The D-isomer of colitose, abequose, is also a candidate for another ligand. TPL2 shows an 80% sequence identity with TL-3 and, like TL-3, specifically interacts with the O-polysaccharide fragment of LPS [123,129,130]. This lectin binds to *E. coli* LPS with a K_D_ of 1.03 × 10^−6^ M [131]. The specific ligand for TPL2 is L-rhamnose.

A structural feature of tachylectins is the presence of tandem repeats in their amino acid sequence, which are at least 30 residues long and encode the secondary and tertiary structure of the protein. The TL-1 (27 kDa), TL-2 (27 kDa), and TL-3 (14 kDa) sequences include six, five, and two repeats, respectively [124,125]. TL-3 (14 kDa) is present as a dimer (29 kDa) in solution, while TL-4 (30 kDa) exists under physiological conditions as a high molecular weight oligomer (470 kDa) consisting of 30 kDa subunits [127,128]. Wild-type TPL-2 (18 kDa) exists in solution as a covalent dimer (36 kDa), and the cleavage of the intermolecular disulfide bond results in monomer formation and loss of LPS-binding activity. LPS induces TPL2 oligomerization, in which tetramers and hexamers are formed. In hemolymph plasma, TPL-2 is predominantly present as oligomers with a molecular weight above 60 kDa. Carbohydrate chains of TPL-2 glycoprotein have been suggested to be responsible for the formation of the oligomers’ stable cluster through protein–carbohydrate interactions. Unlike most other LPS-binding proteins, TPL-2 has a near-neutral pI of 7.65. However, there are three clusters of basic amino acids in the TPL-2 sequence that may be critical for its binding to LPS. TPL2 inhibited the growth of Gram-negative *E. coli* but was almost unable to detect Gram-positive bacteria.

The X-ray structures of tachylectin-2 and its complex with N-acetyl-D-glucosamine were solved with a resolution of 2.0 Å [132]. The lectin has a five-bladed β-propeller structure: five four-stranded antiparallel interconnected beta-sheets of W-shaped topology are located around a central water-filled tunnel, with the water molecules arranged as a pentagonal dodecahedron (Figure 8a).

The TL-2 molecule has five equivalent carbohydrate-binding sites located between adjacent β-sheets. The binding sites are formed by a large loop between the outermost strands of β-sheets and the connecting segment of the previous β-sheet (Figure 8b). According to crystal structure analysis, TL-1 is the protein of the six-bladed β-propeller structure [133]. The non-covalently bound TL-3 dimer is expected to have a four-blade β-propeller structure. A large number of binding sites in one polypeptide chain convincingly indicates the ability of the lectin to recognize carbohydrate surface structures of pathogens with a sufficiently high density of ligands.

The high affinity and specificity of horseshoe crab lectins binding with a propeller-like fold or oligomeric organization to a ligand is achieved due to their multivalence, short distances between individual binding sites (for example, 25 Å and 40 Å for TL-2), and low structural flexibility: upon ligand binding, the conformation of the main or side chains of tachylectins does not change at all. These observations suggest that these lectins can recognize parameters characterizing the distribution of glycan ligands on the cell surface, such as density, mobility and spatial arrangement, and this enables them to distinguish between simple ligands (monosaccharides and oligosaccharides) expressed on both the pathogen and the host and thus to discriminate between self and nonself.

The LPS-binding lectin, which is structurally related to tachylectins, has been found in the marine sponge *Suberites domuncula* [134]. This lectin (27 kDa), like TL-1, has six tandem repeats of 30–38 amino acid residues in sequence and exhibits high activity against Gram-negative bacteria, which is inhibited by D-GlcNAc, but not by D-GlcN. A number of proteins of the LPS- and β-1,3-glucan-binding proteins (LGBP) family can be assigned to lectins interacting with LPS. Many of them show a high degree of homology with invertebrate 1,3-β-glucanases. Probably, during the evolution, one of the gene copies of this enzyme evolved towards the specialization of the encoded protein exclusively for the function of binding to LPS and 1,3-β-glucans as a trigger mechanism for the body’s defense response to the introduction of pathogenic microorganisms. At the same time, the protein lost its enzymatic activity, but its catalytic domain became a binding site not only for β-1,3-glucan but also for LPS [135]. Proteins of this family have been most extensively studied in crustaceans [136,137,138]. Recombinant LGBP (PmLGBP) from the *P. monodon* shrimp with a calculated molecular weight and pI of 39.8 kDa and 4.28, respectively, binds LPS with the apparent dissociation constant of 3.55 × 10^−7^ M [139]. Strong binding to LPS, as well as agglutinating activity against Gram-negative and Gram-positive bacteria, was shown by rLGBP from the scallop *Chlamys farreri* [140].

## 8. Conclusions

The present literature review showed that the LPS-binding proteins from marine invertebrates are understudied, which is confirmed by the data summarized in Table 1. These proteins have been characterized in a small number of invertebrate species that mainly inhabit the tropical seas of Southeast Asia and are objects of mariculture in the countries of the region. In addition, the study of ILBPs from new species is often limited by the search for homologues of already-known proteins. Basically, the field of researchers’ interest is focused on the antimicrobial properties of these proteins, while their potential LPS-binding and LPS-neutralizing activities remain unidentified. Currently, it is impossible to exclude the discovery of new ILBP structural types under their targeted search involving new species of marine invertebrates. So a recent screening of marine invertebrates from the Sea of Okhotsk belonging to different taxonomic groups revealed a large number of species with LPS-binding activity that are of interest as new sources of ILBPs [141].

It is noteworthy that most of the well-studied ILBPs were originally isolated from the hemolymph of horseshoe crabs (lat. Xiphosura). This is for several reasons. The horseshoe crabs have the best-characterized immune system of any long-lived invertebrate. The study of immunity in the horseshoe crab has been facilitated by the availability and ease of collecting large volumes of blood. In addition, these marine animals have existed on Earth for about half a billion years and are often referred to as living fossils. The horseshoe crab habitat is rich in pathogenic microorganisms and contains a vast amount of endotoxin, since most aquatic bacteria are of the Gram-negative variety. During their long evolution, horseshoe crabs have formed a unique and very efficient host defense system, which includes a large set of proteins and peptides with high antimicrobial and endotoxin neutralizing activity. These defense molecules are attracting much attention from researchers as potential therapeutic agents.

Endotoxin-neutralizing ILBPs exhibit some structural features that provide optimal parameters for their interaction with LPS and inhibition of LPS toxicity. These proteins/peptides are mostly cationic amphiphilic molecules that have clusters of hydrophobic and hydrophilic amino acid residues on their surface [41,142]. Positively charged residues play a key role in binding ILBP to LPS and neutralizing endotoxin. Anionic ALF peptides that have lost most of these residues in the LPS-binding domains are unable to interact with LPS and exhibit low antimicrobial activity [143]. A high positive charge allows ILBPs to replace divalent cations, approach LPS molecules through strong electrostatic interactions, and neutralize and even overcompensate their negative charge. It could be argued, however, that the proper positioning of the basic amino acids in the three-dimensional structure of the protein is more important than the overall basic (cationic) nature of the protein for binding to the negatively charged LPS. An important factor is the distance between the positively charged residues in the ILBPs. The charged amino groups of Arg and Lys in ILBPs bound to LPS micelles show a typical distance range of 12–15 Å, which is in good accordance with an average distance between phosphate groups in lipid A [144,145]. This fact may mean that the positively charged amino acids residues in ILBPs mainly interact with the lipid A phosphate groups.

The presence of hydrophobic residues in the molecule, along with cationic ones, allows ILBPs to penetrate deeply into LPS micelles and bilayers and interact with lipid A acyl chains. A strong positive correlation is observed between hydrophobicity and LPS-binding activity of ILBPs [146]. An increase in the ratio between hydrophobicity (after reaching a certain threshold) and the net molecule positive charge increases the ability of ILBPs to neutralize LPS. At high hydrophobicity (outside the range), the activity drops, probably due to the strong self-association of ILBPs.

ILBP incorporation into LPS aggregates leads to a change in the endotoxin supramolecular structure [42,147]. The protein-induced conversion of the unilamellar, cubic, or mixed unilamellar/cubic aggregate structures of LPS and lipid A into a multilamellar form is considered as a necessary condition for LPS inactivation. The degree of LPS multilamellarization can directly correlate with the endotoxin-neutralizing activity of ILBPs.

The specific arrangement of amino acid residues in the ILBP molecules is important for the expression of their anti-LPS activity. As illustrated by synthetic peptides (based on the Limulus anti-LPS factor), it was shown that, while they have the same number of cationic and hydrophobic residues at similar sequence positions, they differ from each other through their LPS-neutralizing activity [42]. Moreover, these peptides differ in their ability to neutralize LPS in isolated forms and as constituents of Gram-negative bacteria. Apparently, this fact is explained by the fact that LPS molecules in the aggregate with a cubic structure and in the outer leaflet of the outer membrane of bacteria have different conformations, and peptides with different spatial structures are required for their neutralization. Thus, the geometric correspondence between the ILBP and LPS conformations, which allows positively charged protein residues to bind with high efficiency to the phosphate groups of lipid A, and hydrophobic residues to incorporate into its lipophilic part, can determine the endotoxin-neutralizing activity of the protein. In this regard, a well-organized, stable spatial structure of ILBPs, providing this structural compatibility, may be an important condition for binding lipid A with high affinity, which leads to LPS neutralization. The packaging of aromatic amino acid side chains, perhaps in part because they play an important role in stabilizing the compact structure of ILBPs, has a remarkable impact on the LPS-binding affinity [148]. A significant contribution to the stability of the ILBP structure is made by disulfide bonds. The substitution of cysteine residues in ALF, accompanied by the removal of a disulfide bond, can lead to a loss of endotoxin-neutralizing activity [40]. At the same time, the fully unfolded analog of tachyplesin-1, which has lost disulfide bonds, acquires a well-ordered structure upon binding to the LPS bilayer [144]. Moreover, ILBPs, which are mainly unstructured in solution, can gain an ordered conformation upon interaction with LPS micelles.

The enhancement of LPS-binding and -neutralizing activities of ILBP can be achieved by creating tandem repeats of the LPS-binding units in its molecule or by forming oligomeric forms of the protein. The effectiveness of this multivalent strategy for an improvement in the activity is demonstrated by ILBPs such as Factor C and tachyplesins [97,123].

Although ILBPs remain one of the most promising molecules for the development of endotoxin-neutralizing drugs, there are serious limitations to their introduction into medical practice. They can be unstable under physiological conditions (in particular, they are attacked by proteolytic enzymes) and toxic to mammalian cells and are quickly eliminated from the body [149]. The effective neutralization of LPS requires high therapeutic concentrations of ILBP, which causes serious side effects. To address these shortcomings, the structure of ILBPs is modified, or their synthetic derivatives are obtained. Thus, the stability of LBPs and their resistance to enzymatic degradation can be increased by the cyclization of the peptide (linking the C- and N-terminus), the introduction of D-isomers or unnatural amino acids into the peptide sequence, and their association with nanoparticles [150]. Covalent binding to polyethylene glycol increases the bioavailability of LBPs due to a decrease in the rate of renal clearance [151]. High manufacturing costs represent another major challenge for therapeutic applications of ILBPs. The development of recombinant DNA technologies and the solid-phase peptide synthesis method (SPPS) will contribute to solving this problem.

Synthetic peptides based on the structure of LPS-binding domains of known natural endotoxin-neutralizing ILBPs are of great interest as potential drugs for the treatment of sepsis [42]. Such peptides differ from their natural counterparts in size, amino acid substitutions, and other structural modifications introduced to increase their potential pharmacological efficacy and safety and considering the relationship between structure and biological activity established in the study of native ILBPs. The designed peptides demonstrate significant protective effect against septic shock in animal sepsis models even at a low peptide dose, a rather long half-life, low cytotoxic, and hemolytic activity.

However, ILBP molecules that show high efficacy in the treatment of sepsis in laboratory animals have been unsuccessful in human trials. This can partly be explained by the incomplete adequacy of mouse models of sepsis and septic shock [152]. Another reason may be the variability in the pathogenesis of various septic complications and the heterogeneity of patients, which must be taken into account when developing a strategy for clinical trials. For a correct assessment of the effectiveness of potential anti-endotoxic drugs, certain conditions must be met during trials: clinically proven selection of a suitable type of patients; the early recognition of sepsis and well-timed initiation of goal-directed therapy, which can interrupt the inflammatory cascade, preventing the progression to septic shock with multiple organ dysfunction; and the optimal duration of therapy.

Despite all the difficulties and disappointments, ILBPs remain one of the most promising molecules that can effectively neutralize bacterial endotoxins and inhibit the development of a systemic inflammatory response with cytokine overproduction (a cytokine storm). These molecules are often multifunctional [153]. In addition to the antiendotoxic activity, many peptides may exhibit antimicrobial and multifaceted immunomodulatory properties, possibly resulting in their wider therapeutic possibility. However, further research is required to evaluate the potential of LPS-neutralizing molecules with additional beneficial properties in the treatment of sepsis.

## Figures and Tables

**Figure 1 marinedrugs-21-00581-f001:**
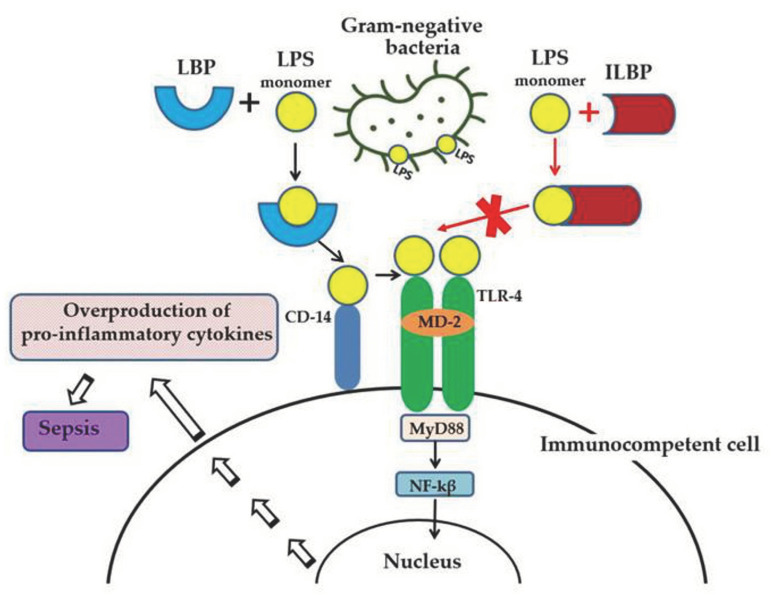
LPS-induced inflammatory response of the innate immune system and the anti-inflammatory effect of LPS-binding peptides/proteins from marine invertebrates. Serum protein LBP (LPS-binding protein) binds the monomer of LPS and delivers it to a CD14 molecule. CD14 transfers LPS to the ectodomain of the TLR4/MD-2 receptor complex, which leads to homodimerization of TLR4. This change in TLR4 conformation provides a binding site for adaptor molecule MyD88 (myeloid differentiation primary-response protein 88). The MyD88-dependent signaling pathway leads to the activation of nuclear factor-κβ (NF-κβ), which regulates the expression of target genes encoding pro-inflammatory mediators. The overproduction of pro-inflammatory cytokines may lead to an uncontrolled inflammatory reaction and eventually to sepsis. The binding of ILBP to LPS blocks CD14–LPS interaction and prevents the transfer of LPS to the TLR4/MD-2 complex, thus interfering with TLR4 dimerization and downstream inflammatory responses.

**Figure 2 marinedrugs-21-00581-f002:**
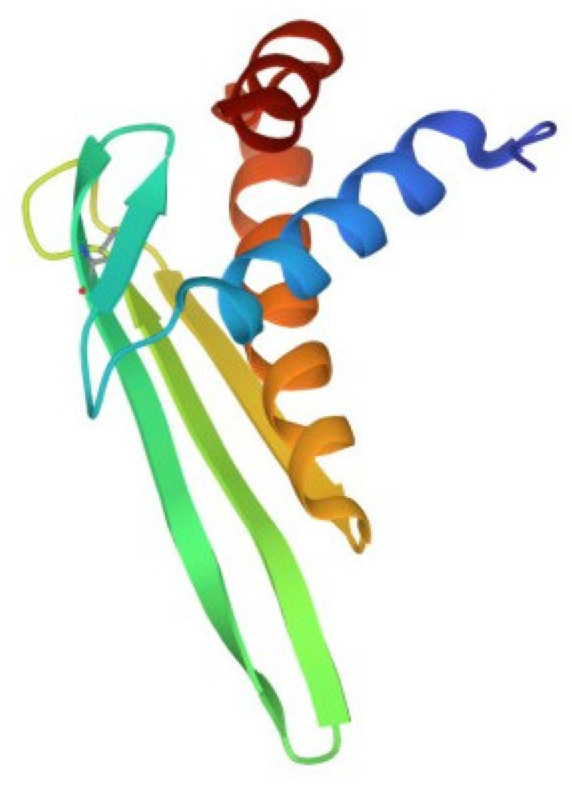
Structure of the anti-lipopolysaccharide factor from *Penaeus monodon* (pdb, 2job).

**Figure 3 marinedrugs-21-00581-f003:**
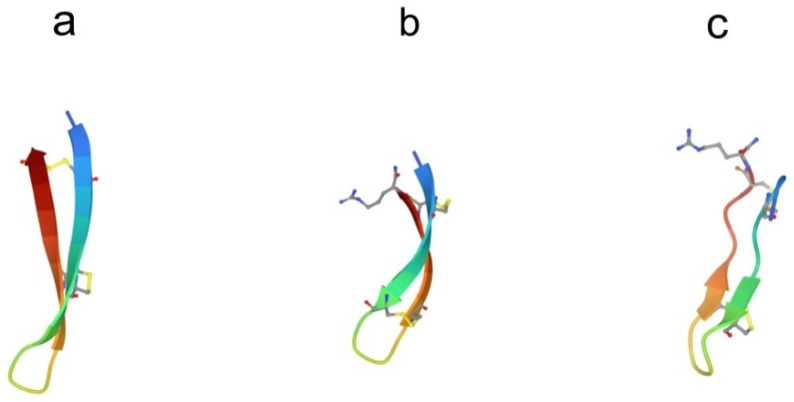
Structure of β-hairpin peptides. (**a**) Arenicin-3 (pdb, 5v0y), (**b**) tachyplesin-1 (pdb,1wo0), and (**c**) polyphemusin-1 (pdb, 1rkk).

**Figure 4 marinedrugs-21-00581-f004:**
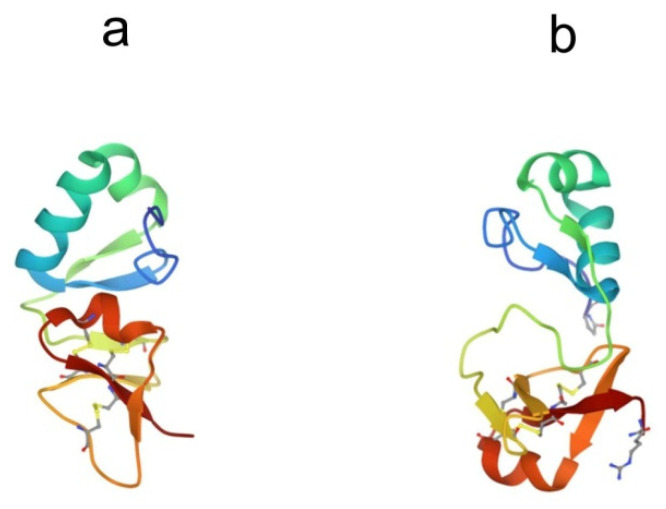
Structures of big defensins isolated from (**a**) horseshoe crab *Tachypleus tridentatus* (pdb, 2rng) and (**b**) oyster *Crassostrea gigas* (pdb, 6qbl).

**Figure 5 marinedrugs-21-00581-f005:**
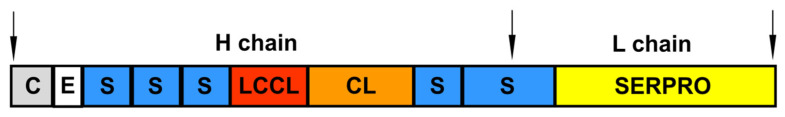
The modular structure of mature Factor C. C—cysteine-rich region (Cys); E—epidermal growth factor (EGF)-like domain; S—complement control protein (CCP) module, also known as Sushi domain; LCCL—domain-type identified in cochlear protein Coch-5b2 and late gestation lung protein Lgl1; CL—C-type lectin-like domain; SERPRO—serine protease domain. The positions of the heavy and light chains are indicated.

**Figure 6 marinedrugs-21-00581-f006:**
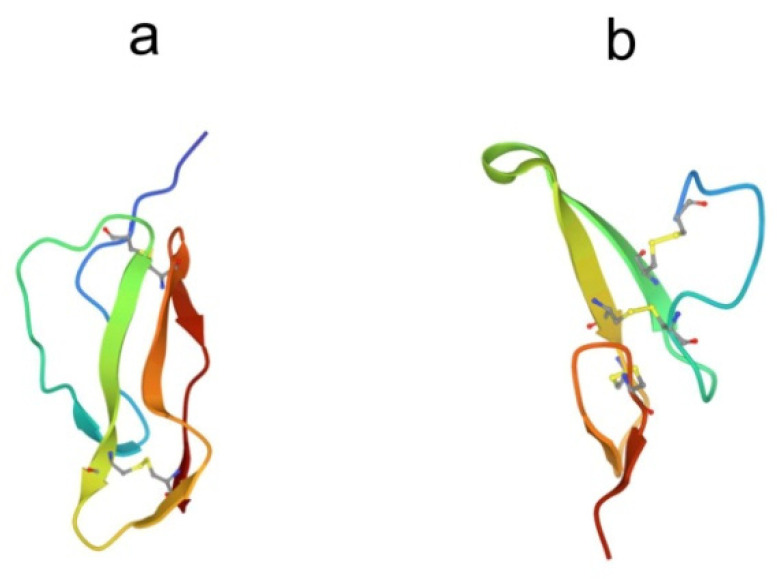
Structures of (**a**) complement control protein module 16 from human complement factor H (pdb, 1hcc) and (**b**) the human epidermal growth factor (pdb, 1jl9).

**Figure 7 marinedrugs-21-00581-f007:**
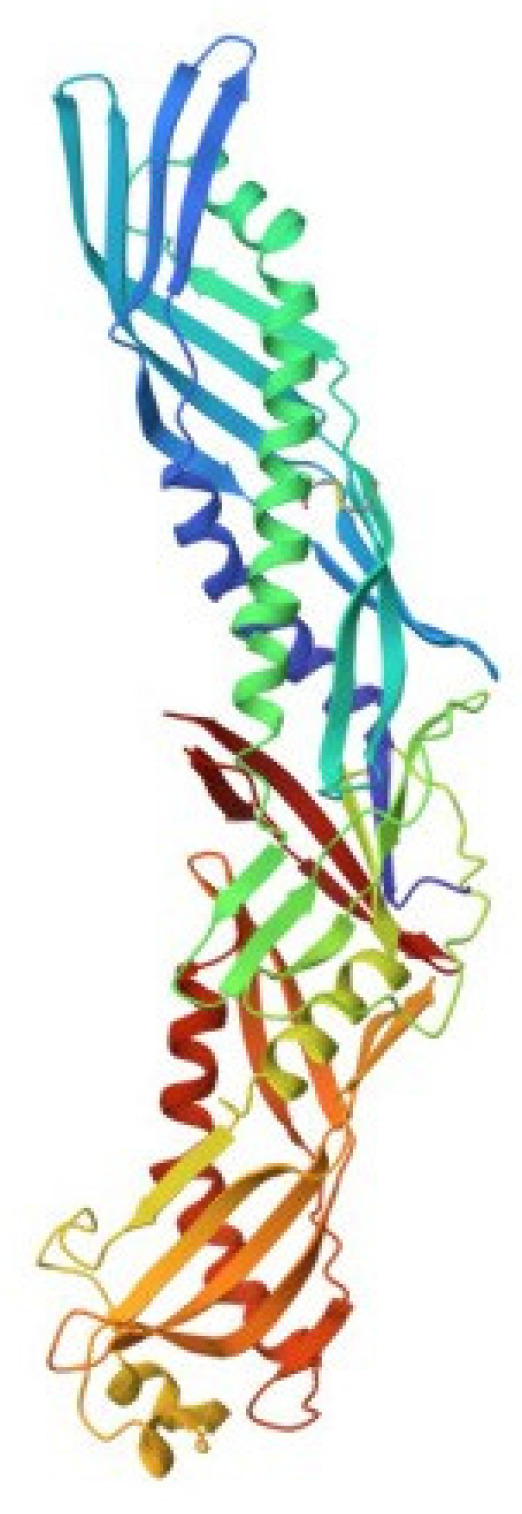
Crystal structure of human bactericidal/permeability-increasing protein (pdb, 1bp1).

**Figure 8 marinedrugs-21-00581-f008:**
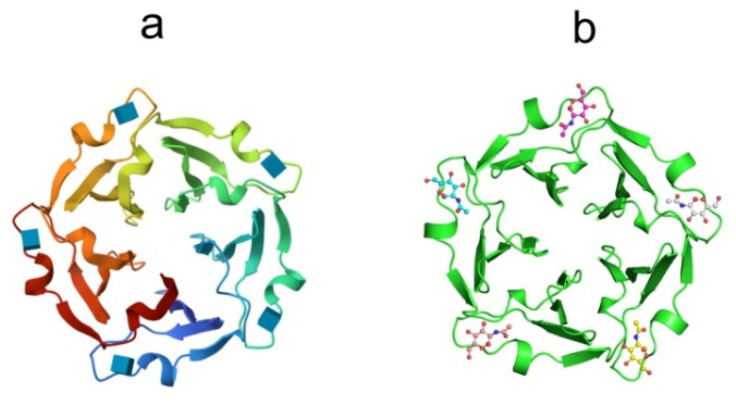
Crystal structures of (**a**) tachylectin-2 (pdb, 1tl2) and (**b**) complex of tachylectin-2 with N-acetyl-D-glucosamine (pdbe, 1tl2). Bound GlcNAc is shown as a ball-and-stick model.

**Table 1 marinedrugs-21-00581-t001:** LPS-binding peptides and proteins from marine invertebrates and their characteristics.

Peptides/Protein	Size, kDa	Structural Characteristics	Biological Activity	Source (Phylum, Species)	Reference
Anti-LPS factor (ALF)	11–12	α-helices, four-stranded β-sheet, disulfide bond	LPS-binding and -neutralization, antibacterial (G−, G+), antiviral, antifungal	Arthropoda—horseshoe crabs *Limulus polyphemus*, *Tachypleus tridentatus*, shrimps *Penaeus monodon*, *Fenneropenaeus chinensis*, *Litopenaeus vannamei*, crabs *Portunus trituberculatus*, *Scylla serrata*, *Eriocheir sinensis, Scylla paramamosai*, lobster *Homarus americanus*	[7,8,9,15,16,17,18,19,20,21,22,35]
Arenicins	2.62	antiparallel β-hairpin, disulfide bonds	anti-endotoxin (optimized arenicin derivatives), antibacterial (G−, G+), antifungal	Annelida—lugworm *Arenicola marina*	[43,47,48]
Tachyplesins, polyphemusins	2.27–2.46	antiparallel β-hairpin, amidated C-terminal arginine residue, disulfide bonds	LPS-binding and -neutralization, antibacterial (G−, G+), antifungal	Arthropoda—horseshoe crabs *L. polyphemus*, *T. tridentatus*, *Tachypleus gigas*, *Carcinoscorpius rotundicauda*	[53,54,55,56,57,58,59,60,61,62,63]
Big defensins	8–11	α-helices, β-sheets (antiparallel and parallel); two domains, disulfide bonds	LPS- binding; antibacterial (G−, G+), antifungal	Arthropoda—horseshoe crab *T. tridentatus*; Mollusca—oyster *Crassostrea gigas*, clam *Venerupis philippinarum*, scallops *Argopecten irradiant*, *Chlamys nobilis*; Chordata—amphioxus *Branchiostoma japonicum*	[69,70,71,72,73,74,75,76,77,78]
Factor C	120 and 132	β-sheets, disordered segments (loops), multidomain structure, disulfide bonds, tandem modules	LPS-binding (binding sites with K_D_ from 10^−9^ to 10^−10^ M) and -neutralization	Arthropoda—horseshoe crabs *L. polyphemus*, *T. tridentatus, Carcinoscorpius rotundicauda*	[80,81,86,87,88,96,97]
Bactericidal/permeability-increasing proteins (BPI)	50.1	α-helix, β-sheet; two-domain “boomerang-like” structure, disulfide bonds	LPS-binding and -neutralization; antibacterial (G−)	Mollusca—oyster *Crassostrea gigas*, squid *Euprymna scolopes*; Annelida—worm *Platynereis dumerilii*; Echinodermata—urchin *Sterechinus neumayeri*	[110,118,119,120,121]
Tachylectins (TL-1 to TL-4 and TPL2)	27, 27, 14, 30 (TL-1,-2,-3,-4); 18 (TPL-2)	β-sheets (four-stranded antiparallel interconnected), propeller-like fold or oligomeric organization, tandem repeats in sequence	LPS-binding (K_D_ 1.03 × 10^–6^ M for *E. coli* LPS); antibacterial activity (G-)	Arthropoda—horseshoe crab *T. tridentatus*; Porifera—sponge *Suberites domuncula*	[125,126,134]
LPS- and β-1,3-glucan-binding proteins (LGBP)	40-60		LPS-binding (K_D_ 3.55 × 10^−7^ M for *E. coli* LPS); antibacterial (G−, G+)	Arthropoda—shrimps *Penaeus monodon*, *Fenneropenaeus merguiensis*, crab *Eriocheir sinensis*; Mollusca—scallop *Chlamys farreri*	[136,137,139,140]

G−, Gram-negative bacteria, G+, Gram-positive bacteria.

## Data Availability

Protein database (PDB) was used to obtain structural information for the proteins presented in the review.

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
