# Peer review of "Host Defense Proteins and Peptides with Lipopolysaccharide-Binding Activity from Marine Invertebrates and Their Therapeutic Potential in Gram-Negative Sepsis"

_marinedrugs, 2023, doi:10.3390/md21110581_

Round 1

Reviewer 1 Report

Comments and Suggestions for Authors

This draft provides a summary of marine invertebrate-sourced host defense peptides/proteins and their potential in treating LPS-induced sepsis. The authors delve into HDP/HDP classification, structural features, and functioning mechanisms, demonstrating a comprehensive understanding of the literature with fluent language. However, there are some important areas for improvement.

1.     the authors should offer a comprehensive background on the molecular mechanisms of LPS-induced diseases. While the introduction covers upstream LPS-receptor binding events, downstream pathways and effectors are equally crucial and should be included.

2.     in lines 68-71, the statement, “Defense proteins usually recognize conservative biopolymers… In Gram-negative bacteria, this biopolymer is LPS,” needs clarification. AMPs can bind various components of Gram-negative bacteria, leading to diverse mechanisms that require further explanation.

3.     Regarding marine invertebrate-sourced peptides, recent literature, including articles from Frontiers in Marine Science (2023;9:2880, doi:10.3389/FMARS.2022.1112595), Nature Product Reports (2023, DOI: https://doi.org/10.1039/D3NP00031A), and Frontiers in Microbiology (2021;12, doi:10.3389/FMICB.2021.785085), should be discussed.

4.     The authors have listed numerous LPS-binding peptides and proteins, indicating their strong binding activity. To support this claim, more detailed quantification data, such as a table summary, should be included.

5.     in line 469, the statement, “However, this fact only applies to those LPS that have carbohydrate motifs recognized by the lectins,” lacks a supporting reference. A proper citation should be added to validate this statement.

6.     lectins are known for their hemocyte agglutination effect and immunomodulatory effect, which should be mentioned in the context of their toxicity.

Comments on the Quality of English Language

see comments

Author Response

  1. the authors should offer a comprehensive background on the molecular mechanisms of LPS-induced diseases. While the introduction covers upstream LPS-receptor binding events, downstream pathways and effectors are equally crucial and should be included.

Figure 1 is included in the text of the manuscript. The caption below to the figure provides the required information about the mechanism of the LPS-induced inflammatory response in the host.

  1. in lines 68-71, the statement, “Defense proteins usually recognize conservative biopolymers… In Gram-negative bacteria, this biopolymer is LPS,” needs clarification. AMPs can bind various components of Gram-negative bacteria, leading to diverse mechanisms that require further explanation.

Llipopolysaccharide (LPS) from Gram-negative bacteria is ubiquitous and the most-studied pathogen-associated molecular pattern (PAMP). These PAMPs allow the innate immune system to recognize pathogens and thus, protect the host from infection. Changes and clarifications have been made to the text.

  1. Regarding marine invertebrate-sourced peptides, recent literature, including articles from Frontiers in Marine Science (2023;9:2880, doi:10.3389/FMARS.2022.1112595), Nature Product Reports (2023, DOI: https://doi.org/10.1039/D3NP00031A), and Frontiers in Microbiology (2021;12, doi:10.3389/FMICB.2021.785085), should be discussed.

The literature contains a number of reviews that summarize information on antimicrobial peptides (AMPs) from marine animals. However, the authors are interested in AMPs only as compounds with antimicrobial activity and potential new antibiotics. The ability of some AMPs to bind and neutralize LPS is not discussed in any way.

  1. The authors have listed numerous LPS-binding peptides and proteins, indicating their strong binding activity. To support this claim, more detailed quantification data, such as a table summary, should be included.

Quantitative data characterizing the binding affinity of ILBP to LPS (dissociation constants), if determined, are given in the text and in the summary table, which we included in the manuscript at the suggestion of the reviewers.

  1. in line 469, the statement, “However, this fact only applies to those LPS that have carbohydrate motifs recognized by the lectins,” lacks a supporting reference. A proper citation should be added to validate this statement.

This statement, as it seems to us, is based on an obvious fact. If there are no carbohydrate structures on the LPS molecule corresponding to the carbohydrate-binding sites of lectins, the LPS-lectin interaction cannot take place. As known, carbohydrate ligands in the LPS molecule are localized mainly in O-specific polysaccharide chains. LPS from different bacteria differ in the structure of this hyper variable carbohydrate fragment and, therefore, in their ability to bind to a specific lectin.

  1. lectins are known for their hemocyte agglutination effect and immunomodulatory effect, which should be mentioned in the context of their toxicity.

Host defense proteins have been shown to be polyfunctional compounds.  They not only have a strong antimicrobial effect, but are also involved in immune regulation, antitumor effects, and other function. However, further research is required to evaluate the potential of these molecules with additional beneficial properties when used as drugs. In addition, they can be toxic to mammalian cells. In this regard, lectins are no exception. We discuss these issues in the “Conclusion” section.

Reviewer 2 Report

Comments and Suggestions for Authors

The work by Solov’eva et al in this research explores a critical and timely topic - sepsis, which is a life-threatening condition often resulting from the excessive and uncontrolled activation of the immune system in response to infection. The role of lipopolysaccharide (LPS), also known as endotoxin, in the development of Gram-negative sepsis is well-established, and this review focuses on a promising avenue for treatment. One of the strengths of this work is its focus on LPS-binding proteins from marine invertebrates (ILBPs) as potential agents for sepsis treatment. This unique approach draws attention to nature's own solutions and could hold promise for the development of effective treatments where none currently exist. The exploration of the structure, physicochemical properties, antimicrobial abilities, and LPS-binding/neutralizing activity of these proteins and their synthetic analogues is commendable, as it provides a comprehensive understanding of their potential as therapeutic agents. Additionally, the inclusion of a discussion on the challenges faced during clinical trials of potential anti-endotoxic drugs adds a practical dimension to the research. Clinical trials are a crucial step in bringing potential treatments to patients, and understanding the issues that may arise is vital for translational research. I have the following comments on the work, need to address.

1.       Sequence and structural relationship: Is there any relationship between the primary sequences of the lipopolysaccharides binding proteins and peptides from Marine invertebrates? This part should be analyzed, and results should be included.

2.       One master figure expining the whole concept of the review article should be added.

3.       The tabular representation of the peptide/proteins would be useful for the reader.

4.       Existing data about the abundance of occurrence of lipopolysaccharide binding peptides/proteins in various marine invertebrates should be outlined and discussed.

Author Response

  1. Sequence and structural relationship: Is there any relationship between the primary sequences of the lipopolysaccharides binding proteins and peptides from Marine invertebrates? This part should be analyzed, and results should be included.

Lipopolysaccharide-binding proteins/peptides belonging to different groups do not show similarities in their primary sequences. However, these molecules have the same structural motifs that ensure their interaction with LPS and inhibition of its toxic effects. These structural features of ILBP are discussed in the Conclusion section. As known, the exact identity of amino acids is not critical for maintaining the overall fold of a protein. In recent years, many examples of structurally similar proteins with undetectable sequence similarities have been identified [Zang et al., 1997].

  1. One master figure expining the whole concept of the review article should be added.

Figure 1 is included in the text of the manuscript.

  1. The tabular representation of the peptide/proteins would be useful for the reader.

A table 1 summarizing the data on peptides/proteins presented in the review has been added to the textof the manuscript.

  1. Existing data about the abundance of occurrence of lipopolysaccharide binding peptides/proteins in various marine invertebrates should be outlined and discussed. These data are reflected in table1, entered into the text.

These data are presented in table 1, included in the text of the manuscript.ILBPs have been characterized in a small number of invertebrate species that mainly inhabit the tropical seas of Southeast Asia and are objects of mariculture in the countries of the region.

Reviewer 3 Report

Comments and Suggestions for Authors

The review by Solovéva and collaborators provides a comprehensive compilation of intriguing studies characterizing proteins and peptides capable of binding to bacterial lipopolysaccharides. However, several essential aspects must be addressed to meet the standards for publication in Marine Drugs.

Certain paragraphs lack references, such as lines 34 to 36 and 161 to 165. It is imperative to include relevant citations to support the claims made in these sections.

Consider incorporating a summary table detailing peptides and proteins exhibiting LPS binding capacity, along with their respective sources of origin. This table would enhance the clarity and accessibility of the information presented.

The figures included in the manuscript require revisions. They should incorporate specific domains or motifs involved in the binding to LPS. EThis information will significantly improve their contribution to the overall content.

The review emphasizes the conservation levels among different ILBP families. To elucidate this further, it is advisable to conduct a sequence analysis (alignment) of these molecules, considering the diverse invertebrate species in which they have been identified. This analysis will provide valuable insights into the evolutionary relationships and conservation patterns of ILBPs.

To enhance the reader's understanding, consider incorporating a graphical model illustrating the distinct mechanisms of action employed by various ILBP families. A visual representation will facilitate comprehension and comparison of these mechanisms.

Addressing these points will elevate the quality and comprehensiveness of the review.

Author Response

Certain paragraphs lack references, such as lines 34 to 36 and 161 to 165. It is imperative to include relevant citations to support the claims made in these sections.

A reference has been added to the text (lines 34 to 36). A reference [1] contains all the information given in this paragraph (lines 22 to 37). Changes were made to the text according to the comments of the reviewer (lines161 to 165).

Consider incorporating a summary table detailing peptides and proteins exhibiting LPS binding capacity, along with their respective sources of origin. This table would enhance the clarity and accessibility of the information presented.

Summary table 1 has been generated and added to the text.

The figures included in the manuscript require revisions. They should incorporate specific domains or motifs involved in the binding to LPS. This information will significantly improve their contribution to the overall content.

There is no data in the literature on the exact localization of LPS-binding sites on ILBP molecules. This makes it difficult to label these sites in the protein structures shown in the figures. In the case of tachylectin-2, for which this information exists, the binding sites are shown (Figure 7).

The review emphasizes the conservation levels among different ILBP families. To elucidate this further, it is advisable to conduct a sequence analysis (alignment) of these molecules, considering the diverse invertebrate species in which they have been identified. This analysis will provide valuable insights into the evolutionary relationships and conservation patterns of ILBPs.

We agree that conducting such a study may be of interest from the point of view of evolutionary relationships in ILBP. However, this is beyond the scope of this review.

To enhance the reader's understanding, consider incorporating a graphical model illustrating the distinct mechanisms of action employed by various ILBP families. A visual representation will facilitate comprehension and comparison of these mechanisms.

In this review, we consider host defense peptides/proteins that directly bind LPS with high affinity, resulting in inhibition of its interaction with the TLR-4 receptor. Figure 1, added to the text of the manuscript, illustrates this mechanism.